# Invariant Natural Killer T-Cells and Total CD1d Restricted Cells Differentially Influence Lipid Metabolism and Atherosclerosis in Low Density Receptor Deficient Mice

**DOI:** 10.3390/ijms20184566

**Published:** 2019-09-14

**Authors:** Paul A. VanderLaan, Catherine A. Reardon, Veneracion G. Cabana, Chyung-Ru Wang, Godfrey S. Getz

**Affiliations:** 1Department of Pathology, The University of Chicago, Chicago, IL 60637, USA; 2Department of Microbiology and Immunology, Northwestern University, 633 Clark St, Evanston, IL 60208, USA

**Keywords:** NKT cells, atherosclerosis, cholesterol, lipoproteins

## Abstract

Natural killer T (NKT) cells are a distinct subset of lymphocytes that bridge the innate and adaptive immune response and can be divided into type I invariant NKT cells (iNKT) and type II NKT cells. The objective of this study is to examine the effects of NKT cell on lipid metabolism and the initiation and progression of atherosclerosis in LDL receptor deficient (LDLR^−/−^) mice. Mice were fed an atherogenic diet for 4 or 8 weeks and plasma lipids, lipoproteins, and atherosclerosis were measured. The selective absence of iNKT cells in Jα18^−/−^LDLR^−/−^ mice led to an increase in plasma cholesterol levels in female mice. Transgenic Vα14tg/LDLR^−/−^ mice with elevated numbers of iNKT cells had increased late atherosclerosis of the innominate artery, though absence of either iNKT cells or all NKT cells and other CD1d expressing cells had varying effects on atherosclerotic lesion burden in the ascending aortic arch and aortic root. These studies not only highlight the potential modulatory role played by NKT cells in atherosclerosis and lipid metabolism, but also raise the possibility that divergent roles may be played by iNKT and CD1d restricted cells such as type II NKT cells or other CD1d expressing cells.

## 1. Introduction

Atherosclerosis is characterized by the retention and modification of lipids in the vessel wall, initiating a complex, chronic inflammatory reaction that involves both the innate and adaptive immune systems [1,2,3]. Although the adaptive immune system is not obligatory for atherogenesis [4,5,6], it is now well documented that in the normal, fully immune-competent state, both pro- [7,8,9,10] and anti-inflammatory [11,12,13,14,15,16] components of the adaptive immune system can significantly impact atherogenesis. This communication is concerned with the atherogenic role of natural killer T (NKT) cells, so named because they have surface receptors and functional properties of both natural killer (NK) cells of the innate immune system and T-cells of the adaptive immune system [17,18,19,20]. 

NKT cells recognize and respond to lipid or glycolipid antigens in the context of the MHC class I related antigen presenting molecule CD1d [17,18,19,20]. There are multiple forms of CD1 molecules in humans, but mice only express CD1d. There are at least two broad subsets of CD1d restricted NKT cells: those expressing a semi-invariant T-cell receptor (TCR), often referred to as type I or invariant NKT cells (iNKT), which in the mouse express Vα14Jα18 TCR α-chain and Vβ8, -7, or -2 TCR β-chains and NKT cells that express a more diverse set of TCRs and are referred as type II NKT cells [21]. Both sets of NKT cells respond rapidly to engagement of their TCRs with lipid-loaded CD1d by producing a complement of Th1 (e.g., INF-γ and IL-2) and Th2 (e.g., IL-4 and IL-10) cytokines. These cytokines in turn influence many other cells of the innate and adaptive immune systems. Furthermore, NKT cells can demonstrate granzyme mediated cell lysis typical of NK cells, and CD1d independent activation of NKT cells involving Toll-like receptors has been reported [19]. As such, these cells are uniquely positioned as a bridge between the innate and adaptive branches of the immune response. 

Several investigators have studied the role of NKT cells in murine models of atherosclerosis [22,23,24,25,26,27,28,29]. In these studies, the experimental approach has involved either (a) a profound reduction in both subsets of NKT cells accomplished by the genetic elimination of the CD1d molecule, (b) the specific elimination of the iNKT cells via genetic deletion of the Jα18 chain of the invariant TCR, (c) robust activation of iNKT cells with exogenous α-galactosyl ceramide (αGalCer) administration, the physiological significance of which is not clear or (d) adoptive transfer of NKT cells or splenocytes into adaptive immune deficient mice [30]. Each of these studies has been performed in separate laboratories and have employed either wild type [22], apoE deficient (ApoE^−/−^) mice [22,23,24,29], or LDL receptor deficient (LDLR^−/−^) mice [25,26,28]. In only one case have ApoE^−/−^ and LDLR^−/−^ been subjected to the same experimental design in the same laboratory and the atherogenic response to αGalCer mediated activation of NKT cells was distinct for each of these models [27]. In contrast to the other studies, this study was the first to suggest that iNKT cells may be atheroprotective. 

In the current study, we have investigated the effects of an increase or absence of endogenous iNKT cells on both lipoprotein metabolism and atherosclerosis by crossing Vα14 transgenic (Vα14tg) mice and Jα18 deficient (Jα18^−/−^) mice respectively with LDLR^−/−^ mice and feeding the animals an atherogenic Western type diet (WTD). Since type II NKT cells are still present in Jα18^−/−^LDLR^−/−^ mice, we also compare these mice with CD1d^−/−^ LDLR^−/−^ mice in which both subtypes of NKT cells are absent and CD1d expression is lacking on other cell types. In doing so, all of these genetic models were directly compared and studied under identical experimental and dietary conditions in the same laboratory. 

## 2. Results

In this study we examined both male and female mice. However, we noted that in general females were more responsive than males. Consequently, we report the results on female mice in detail referring to the results on male mice when appropriate. In the first section we compare the response of mice with varying levels of iNKT cells (LDLR^−/−^, Vα14tg/LDLR^−/−^, and Jα18^−/−^LDLR^−/−^ mice) to WTD feeding and in the second section we compare the response of mice that lack both iNKT cells and type II NKT cells (CD1d^−/−^LDLR^−/−^ mice) with mice lacking only iNKT cells (Jα18^−/−^LDLR^−/−^ mice).

### 2.1. Animals with Varying Levels of iNKT Cells 

We first discuss the comparison of WTD fed mice with varying numbers of iNKT cells: namely LDLR^−/−^ mice with basal levels of iNKT cells, Vα14tg-LDLR^−/−^ mice with elevated levels of iNKT cells, and Jα18^−/−^LDLR^−/−^ mice lacking iNKT cells (Figure 1). An increased proportion of total splenocytes were iNKT cells in the Vα14tg LDLR^−/−^ mice compared to the wild type LDLR^−/−^ littermates. The Jα18^−/−^LDLR^−/−^ mice lacked iNKT cells, while the CD1d^−/−^LDLR^−/−^ mice completely lacked all CD1d-restricted T-cells including both iNKT cells and non-iNKT cells. As type II NKT cells are present in low abundance and a sulfatide-loaded CD1d tetramer was not available to measure type II NKT cells, we cannot determine the level of this subset in Jα18^−/−^LDLR^−/−^ mice or confirm their absence in CD1d^−/−^LDLR^−/−^ mice. No differences in NKT cell levels were noted between genders. Furthermore, there were no marked alterations in the proportions of other T- or B-cell subsets as assessed by flow in any of these strains (data not shown).

Body weights of the WTD fed mice were similar between the strains (data not shown). The most consistent difference observed in the female mice was an increase of plasma cholesterol levels in the Jα18^−/−^LDLR^−/−^ mice (Figure 2A,B). This increase was reflected in higher VLDL and LDL cholesterol levels (Table 1). There was no difference in VLDL-TG production rates in female Vα14tg/LDLR^−/−^ and Jα18^−/−^LDLR^−/−^ mice (Table 2) suggesting that the higher VLDL levels in the iNKT cell deficient mice may be due to reduced clearance rates. There were only modest differences in plasma lipids in male mice (Appendix A).

To assess the impact of diet on NKT cell levels, we measured NKT cell levels (defined as CD3^+^ αGalCer-loaded CD1d tetramer^+^ cells) in 20 week old Va14tg/LDLR^−/−^ mice maintained on chow or fed WTD for 12 weeks (*n* = 4–7 per group). High-fat diet feeding resulted in a 36% reduction in the proportion of iNKT cells in the spleens of Vα14tg/LDLR^−/−^ mice (from 11.4% ± 1.3% to 7.3% ± 0.4%, *p* < 0.005), suggesting that diet induced hypercholesterolemia itself may lead to a reduction in the number of iNKT cells or induced NKT cell anergy as has been demonstrated by Major and colleagues [31]. As a consequence, cytokine production by the NKT cells may vary through the course of the exposure to the diet-induced hyperlipidemia.

The hepatic lymphocyte pool is characterized by a high frequency of NKT cells [32]. To establish the impact of acute changes in hyperlipidemia on the number or functional status of hepatic NKT cells, we compared NKT cell levels in the liver of female LDLR^−/−^ and Vα14tg/LDLR^−/−^ mice fed WTD for only 3 weeks (*n* = 2–4 per group). Compared to chow-fed mice, WTD feeding led to a 2-fold increase in the number of total lymphocytes in the livers of LDLR^−/−^ mice and a 4-fold increase in the livers of Vα14tg/LDLR^−/−^ mice. In contrast, WTD feeding led to a decrease in the proportion of iNKT cells in the hepatic lymphocyte population in both LDLR^−/−^ mice (from 18% ± 2% to 8% ± 0.7%) and the Vα14tg, LDLR^−/−^ mice (from 33% ± 4% to 22% ± 5%). Although the results were not significantly different, they are consistent with the results in the spleen. However, the WTD feeding did not appear to alter the activation state of the hepatic iNKT cells (Appendix A). The ratio of iNKT cells expressing surface activation markers CD25 and CD69 to iNKT cells with low levels of the activation markers is similar in chow and WTD fed mice. 

The influence of iNKT cells on atherosclerosis appears to be time and vascular site specific. There were no lesions in the innominate artery after 4 weeks of diet, and even at 8 weeks the lesions were very modest (data not shown). After 12 weeks of WTD, innominate artery atherosclerosis was significantly greater in female Vα14tg/LDLR^−/−^ mice compared to LDLR^−/−^ and Jα18^−/−^LDLR^−/−^ mice (Figure 3A). Atherosclerosis along the lesser curvature of the ascending thoracic aortic arch after 4 weeks of WTD was significantly less in Jα18^−/−^LDLR^−/−^ females (Figure 3B), despite higher plasma cholesterol levels (Figure 2A). However, the lesions in the ascending thoracic aortic arch of the Jα18^−/−^LDLR^−/−^ mice was similar to that of the other two strains after 12 weeks on diet. There was no significant difference in lesion sizes in the aortic root between the three groups with varying levels of iNKT cells (LDLR^−/−^, Vα14tg/LDLR^−/−^ and Jα18^−/−^LDLR^−/−^) after 4 or 12 weeks on diet (Figure 3C). These findings suggest that the absence of iNKT cells may influence the initiation of lesions at more distal sites in female mice. Interestingly, increasing or decreasing iNKT cells in male mice had no effect on atherosclerosis at any of the vascular sites examined (Appendix A).

A histopathologic assessment of the atherosclerotic plaques from each of the vascular sites were scored according to the grading scale described in Appendix A. Consistent with the presence of larger lesions, the innominate artery lesions in the Vα14tg/LDLR^−/−^ mice after 12 weeks of diet were more mature than in the other two strains (Table 3). The beginnings of fibrous caps were observed in the innominate artery in the Vα14tg/LDLR^−/−^ mice, while the lesions in Jα18^−/−^LDLR^−/−^ mice and LDLR^−/−^ mice were predominantly foam cell lesions (grading score <2.0). Despite the fact that the ascending aortic arch lesions were larger in the Vα14tg/LDLR^−/−^ and LDLR^−/−^ mice, they were not more mature than the smaller lesions in the Jα18^−/−^LDLR^−/−^ mice. Taken together, our data support the notion that iNKT cells are indeed pro-atherogenic. 

### 2.2. CD1d Expressing and Restricted Cells Have Separate Influences on Lipoprotein Metabolism and Atherosclerosis

CD1d restricts not only iNKT cells but also type II NKT cells and CD1d is expressed on other immune cells and parenchymal cells [33]. Thus, in this section we compare Jα18^−/−^LDLR^−/−^ (lacking only iNKT cells) with CD1d^−/−^LDLR^−/−^ mice (lacking all CD1d expressing and restricted cells). 

In female mice the plasma cholesterol and triglyceride levels of CD1d^−/−^LDLR^−/−^ mice were statistically lower than the Jα18^−/−^LDLR^−/−^mice throughout the diet period (Figure 2C,D), with the exception of the 8-week triglyceride levels. Both VLDL and LDL cholesterol levels in CD1d^−/−^LDLR^−/−^mice were notably reduced compared to the Jα18^−/−^LDLR^−/−^ mice (Table 1). Assessment of hepatic triglyceride secretion in CD1d^−/−^LDLR^−/−^ female mice did not reveal a difference compared to Jα18^−/−^LDLR^−/−^ mice (Table 2). These lipid and lipoprotein differences were not as apparent in the male mice (Appendix A). As a first approximation, this suggests that the type II NKT and/or other CD1d dependent cells, unbalanced by iNKT cells, may play an important role in elevating plasma apoB containing lipoproteins particularly in female mice. 

As with iNKT cells, the absence of all CD1d-restricted and expressing cells in the CD1d^−/−^ mice exerts a site-specific effect on atherosclerosis (Figure 3). At 12 weeks of diet feeding, there were no differences in atherosclerosis in the innominate artery. On the other hand, in the ascending aortic arch, the early lesions after 4 weeks of diet were significantly smaller in the Jα18^−/−^LDLR^−/−^ mice (Figure 3B). However, after 12 weeks of diet the opposite was observed with significantly smaller ascending aortic arch lesions observed in CD1d^−/−^LDLR^−/−^ mice than in the Jα18^−/−^LDLR^−/−^ mice. This effect was also observed in male mice (Appendix A). Indeed, between 4 and 12 weeks of diet the rate of lesion growth in the ascending aortic arch of the Jα18^−/−^LDLR^−/−^ mice was four times faster than in CD1d^−/−^LDLR^−/−^ mice. This argues that, at least at this arterial site, CD1d-restricted or expressing cells other than iNKT cells play an overall proatherogenic role and in their absence, progression of atherosclerosis is slowed.

Aortic root lesions in the female CD1d^−/−^LDLR^−/−^ mice were significantly larger than in the Jα18^−/−^LDLR^−/−^ mice after 4 weeks of diet (Figure 3C). Similar results were obtained in male mice (Appendix A). The increased early atherosclerosis in the aortic root of the CD1d^−/−^LDLR^−/−^ mice was surprising since the Jα18^−/−^LDLR^−/−^ have similar levels of atherosclerosis as the LDLR^−/−^ and Aslanian and colleagues [25] reported a significant decrease in early aortic root atherosclerosis in CD1d^−/−^LDLR^−/−^ mice compared to LDLR^−/−^ mice. There were differences in the cholesterol content of the diets used in the two studies (0.2% in this study vs. 1.25% in the Aslanian study). However, when our animals were fed the 1.25% cholesterol diet, the LDLR^−/−^ and CD1d^−/−^LDLR^−/−^ mice had similarly sized aortic root lesions in the aortic root (Appendix A). However, the LDLR^−/−^ mice fed the 1.25% cholesterol diet had slightly larger lesions than the same animals fed the 0.2% cholesterol diet (compare Appendix A and Figure 3C). Although not explaining the difference between our results and those of Aslanian and colleagues, they do indicate that this early lesion may be quite sensitive to the diet composition.

### 2.3. Hepatic Gene Expression.

In an attempt to explain the differences in plasma lipids and lipoproteins, hepatic gene expression was analyzed on female mice after 12 weeks of WTD feeding (Appendix A and Appendix A). While a few differences were observed for genes involved in lipoprotein metabolism and cytokines expressed by NKT cells, none of these differences are able to account for the differences in plasma lipid levels.

## 3. Discussion.

We demonstrate that variations in NKT cell populations or other CD1d expressing cells influence plasma lipids and lipoproteins and selectively affect atherosclerosis at three vascular sites in a site and time dependent fashion. Of particular interest is the observation that there are notable differences between the selective effects of iNKT cell deficiency (in Jα18^−/−^LDLR^−/−^ mice) as compared to absence of all CD1d restricted T-cells and CD1d expressing cells (CD1d^−/−^LDLR^−/−^ mice), highlighting the possibility that the type II NKT cells or other CD1d expressing cells may be active in lipoprotein homeostasis and in atherogenesis. We recently reported that these male Jα18^−/−^ LDLR^−/−^ mice fed an obesogenic high sucrose and high cholesterol diet at the University of Washington had increased atherosclerosis compared to LDLR^−/−^ and CD1d^−/−^LDLR^−/−^ mice [34]. These two studies are the first reports in which Jα18^−/−^LDLR^−/−^ and CD1d^−/−^LDLR^−/−^ mice have been examined alongside one another in a comparable experimental protocol.

In this study, we relied on the NKT cells responding to endogenous antigens presumably generated as a result of the diet-induced hypercholesterolemia. We have previously shown that an NKT cell activating antigen(s) is present in the plasma of LDLR^−/−^ mice fed a WTD and is absent in the plasma of ApoE^−/−^ mice [28]. The nature of this endogenous antigen is unknown, although it is assumed to be a glycolipid [17,18,19,20].

### 3.1. NKT Cell Status and Lipoprotein Homeostasis

We observed little difference in plasma lipids in the presence of an overabundance of iNKT cells (Vα14tg mice). This suggests that there is not a clear relationship between number of hepatic NKT cells and the lipoprotein phenotype. On the other hand, the selective absence of iNKT cells in Jα18^−/−^LDLR^−/−^ mice was associated with a notable increase in plasma lipid levels, involving increases in VLDL and LDL. Similar increase was observed previously in male Jα18^−/−^LDLR^−/−^ mice fed the obesogenic diet [34]. This increase in lipoprotein levels could be attributable to a direct suppressive effect of iNKT cells on lipoprotein homeostasis. Alternatively, this increase may be the result of an imbalance in the interaction between iNKT cells and other CD1d-expressing or restricted cells including type II NKT cells, such that in the absence of iNKT cells, the influence of these other cells would be unopposed. The precise mechanism(s) by which these CD1d-expressing or restricted cells influence lipoprotein metabolism is not clear but does not appear to be attributable to changes in VLDL-triglyceride production by the liver. In the only other study of Jα18^−/−^LDLR^−/−^ mice, no differences in plasma lipids or lipoproteins were seen after 8 weeks of WTD compared to LDLR^−/−^ mice [26]. In our current study changes were noted only in female mice, not males. It is furthermore important to note that the NKT cells in the liver represent a dynamic population of cells, declining in pool size with the duration of hypercholesterolemia. As with atherosclerosis, there are several parameters that influence the lipoprotein phenotype of the mice, particularly the nature and duration of diet and gender.

### 3.2. NKT Cell Status and Atherosclerosis Phenotype

Our atherosclerosis results are characterized by site selective effects. The major differences in atherosclerosis were seen in the group lacking only iNKT cells (Jα18^−/−^ LDLR^−/−^ mice). The lesions at early time points in the ascending aortic arch were smaller in this strain, despite having the highest plasma lipid levels. This is consistent with a previous study of Jα18^−/−^LDLR^−/−^ mice fed WTD for 8 weeks [26]. Although lesion initiation was slower, the rate of progression, especially at the ascending aortic arch, was more rapid (Figure 3, compare 4–12-week differences). This could be attributable to the high levels of apoB-lipoproteins. 

It should be noted that all studies of Jα18^−/−^ mice used the same strain in which a neomycin resistant gene under the control of the PGK promoter was inserted into the *Traj18* loci that encodes the TCRα-chain region 18 that together with the Vα14 variable region generates the TCRα chain on iNKT cells with the appropriate antigen specificity [35]. However, it has recently been shown that the use of Jα regions upstream of Traj18 was significantly reduced in these knockout mice resulting in a loss of the diversity of the TCRα repertoire [36]. This includes the Jα33 region essential for the development of another innate-like T-cells, the mucosal-associated invariant T-cells (MAIT) [37]. Whether MAIT cells impact on lipid metabolism or atherosclerosis has not been examined. 

The absence of all CD1d-expressing and restricted cells including type II NKT cells in addition to iNKT cells also resulted in complex effects on atherogenesis. Early lesion formation in the aortic root and ascending aortic arch was more rapid in both female and male CD1d^−/−^LDLR^−/−^ mice compared to Jα18^−/−^LDLR^−/−^ mice. However, as the lesions matured and grew in size, the anti-atherogenic effect of the absence of both NKT cell subsets and other CD1d expressing cells on atherosclerosis at the aortic root was loss and led to increased atherosclerosis at the ascending aortic arch. Similar atherosclerosis results were obtained in the male mice despite no difference in plasma lipids, suggesting these atherosclerosis effects in the female mice are likely not simply due to alterations in plasma lipid levels. While there is no method available to ablate type II cells, a recent meeting abstract claims that even in the absence of CD1d type II NKT cells are present in the lungs based on sulfatide loaded CD1d tetramer labeling [38]. However, the cells did not appear to be functional. This observation needs further study. 

Cross talk between iNKT cells and type II NKT cells has been described in relation to tumor immunity and tumor surveillance and in relation to parasitic infection [39,40]. The opposite effects on the initiation of atherosclerosis, particularly in the aortic root, independent of lipid levels, argues for a possible influence of NKT cell-type balance on events associated with lesion initiation, such as activation of the endothelium, monocyte adhesion, and emigration of monocytes into the intima. Imbalances in NKT cell subtypes is thought to influence some autoimmune disorders and antitumor immunity [40]. If the atherosclerosis effects are attributable to an altered balance between the iNKT cells and type II NKT cells, the cells must secrete quite powerful mediators, as they represent quite a small proportion of total NKT cell population. In addition, to account for the different patterns of lesion initiation and progression, their relative contribution likely changes throughout the course of the experiment. Furthermore, the NKT cell dependent immunoregulatory network may influence the other cells of the immune system known to affect atherosclerosis [41]. For example, it has been shown that the acute phase isoforms of serum amyloid A (SAA), which is increased upon feeding LDLR^−/−^ mice a Western type diet [42], promotes the interaction between NKT cells and IL-10 producing neutrophils, leading to attenuation of the expression of this anti-inflammatory cytokine [43].

Several of the previous studies of NKT cells have been performed with ApoE^−/−^ mice [22,23,24,27]. It has been suggested that apoE may be involved in endogenous antigen presentation to activate NKT cells [44], so this model may confound analysis of NKT cell involvement in atherosclerosis. In several of these studies the superagonist αGalCer was used. This agonist may profoundly affect the cytokine milieu [23], so it is not clear that this approach reflects the physiological circumstance involving endogenous activation and modulation of NKT cell behavior. We have shown that the adoptive transfer of Vα14 transgenic splenocytes into RAG^−/−^ LDLR^−/−^ recipients promotes atherosclerosis in the aortic root [28]. In the current study, increasing iNKT cells lead to an increase in innominate artery atherosclerosis, but only in female mice. While both studies observed a pro-atherogenic effect of iNKT cells, different arterial sites were affected. This could be due to site-specific interactions between iNKT cells and other T-cells or B-cells that are absent in the RAG^−/−^ LDLR^−/−^ recipients in the adoptive transfer study. 

As mentioned previously, our results on the CD1d^−/−^LDLR^−/−^ mice are in contrast to Aslanian and colleagues [24]. We ruled out differences in diet as a factor contributing to the different results. It is also worth noting that the CD1d^−/−^ mice used in the two studies were generated in different laboratories. Mice contain two CD1d genes, *Cd1d1* and *Cd1d2* [45]. Recent studies have shown that the two CD1d molecules present different self-antigen molecules in the thymus thus leading to iNKT cells with different T-cell antigen receptor repertoire [46]. The Aslanian study employed animals from Luc Van Kaer in which only the *Cd1d1* gene is eliminated [47] while our studies were performed with animals from Chyung-Ru Wang [48] in which both genes were knocked out. It remains to be seen whether differences among these two independently derived CD1d^−/−^ mice including differences in subsets of NKT cells contributed to the differing atherosclerosis results. 

The CD1d influence on atherosclerosis and lipid metabolism may be a consequence of CD1d expression on parenchymal cells, such as liver or intestinal cells [33,49,50]. In a prior study we observed an increase in plasma cholesterol following the adoptive transfer of CD1d deficient splenocytes to RAG^−/−^ LDLR^−/−^ mice compared to the adoptive transfer of wild type splenocytes [28]. In this study, CD1d was expressed on non-immune cells. It should be noted that CD1d is highly expressed on B regulatory cells that express IL-10 [51]. Thus, we need to entertain the possibility that the absence or reduced function of B regulatory cells or other CD1d expressing cells could play a role in the early increment in aortic root atherosclerosis seen in the CD1d deficient mice. We have preliminary data suggesting that in CD1d^−/−^LDLR^−/−^ mice there is a two-fold reduction in IL-10 producing B-cells. In addition, it has recently been shown that disruption of the interaction of CD1d on mast cells with NKT cells aggravates atherosclerosis [52].

One possible explanation of the difference between our results with CD1d^−/−^ LDLR ^−/−^ mice and those reported previously is that the studies were performed in different vivaria likely populated by different intestinal microbiomes. It is well established that there is a complex interaction between the intestinal microbiome and the phenotype of NKT cells in the intestine [53], which may in turn influence the processes of lipoprotein regulation and atherogenesis. In particular the Paneth cells of the intestine express CD1d and release antimicrobial peptides in a CD1d dependent fashion [54]. This could be one of the determinants of atherogenesis in the model of CD1d deficiency perhaps independent of the direct involvement of NKT cells.

## 4. Materials and Methods

### 4.1. Mice 

The mice were housed in specific pathogen free barrier facilities and experimental procedures performed in accordance with National Institutes of Health guidelines under protocols approved by the University of Chicago Institutional Animal Care and Use Committee (ACUP 69271, approved July 2017). All mice were backcrossed into the C57BL/6 genetic background for at least 10 generations. The Vα14Jα18 TCR transgenic mice and the Jα18^−/−^ mice were kindly provided by Albert Bendelac from the University of Chicago [35,55]. The CD1d^−/−^ mice were generated and characterized in the laboratory of Chyung-Ru Wang [48]. The mice were crossed with LDLR^−/−^ mice (Jackson Laboratories) to obtain double knockout mice or LDLR^−/−^ mice hemizygous for the Vα14 transgene. The non-transgenic littermates were used as the LDLR^−/−^ control mice in these studies. All groups of mice bred well with normal sized litters and healthy pups without any apparent abnormalities. At 8 weeks of age, the mice were switched from a standard laboratory chow diet (Harlan Teklad TD7913, 6.25% fat) to the high fat WTD (Harlan Teklad TD88137, 21% saturated fat, 0.15% cholesterol). Bleeds via the retro-orbital sinus were performed on 4 h fasted mice on a monthly basis to determine fasting plasma cholesterol and triglyceride levels using kits from Roche Diagnostics. At sacrifice, plasma lipoprotein levels were determined by fractionation using tandem Superose 6 FPLC columns from Amersham Biosciences as described previously [56]. 

### 4.2. Lymphocyte Isolation 

Single cell suspensions of splenic lymphocytes were prepared using a Lympholyte-M (Cedarlane Laboratories, Burlington, ON, Canada) density gradient. Single cell suspensions of hepatic mononuclear cells were prepared using a Percoll (Sigma, St. Louis, MO, USA) density gradient [57]. Resident peritoneal cells were obtained by lavaging the peritoneal cavity with sterile phosphate buffered saline containing 2% fetal bovine serum.

### 4.3. Flow Cytometery 

Aliquots containing 10^6^ cells were FcγR blocked (2.4G2) and stained with fluorescently labeled anti-CD3 (17A2), NK1.1 (PK136), CD19 (1D3), CD23 (B3B4), CD5 (53-7.3), CD11b/Mac-1 (M1/70), CD14 (Sa2-8), CD69 (H1.2F3), CD25 (PC61.5), and γδ TCR (GL3) (all from eBioscience, San Diego, CA USA), and αGalCer loaded CD1d-tetramers (mCD1d/PBS-57; NIH MHC Tetramer Core Facility, Atlanta, GA, USA). Flow cytometry was performed using a FACSCalibur (BD Biosciences, San Jose, CA, USA) with subsequent data analysis using WinMDI 2.9 software (Purdue University Cytometry Laboratories, West Lafayette, IN, USA).

### 4.4. Tissue Preparation and Histomorphology 

After 4, 8, or 12 weeks of WTD feeding, the mice were anesthetized, exsanguinated, arterial vasculatures perfused with a paraformaldehyde solution under physiologic pressures, and the upper aortic vasculature tissue mounted in OCT as described previously [28,56]. Serial 10 µm frozen sections were collected from the innominate artery through the aortic root. Lesions in the innominate artery were quantified as the average of three oil red O stained sections separated by 100 µm and located between 100 and 300 µm distal to the apex of the lesser curvature of the aortic arch. Ascending thoracic aortic arch lesions were assessed as the average of three sections separated by 100 µm located between 100 and 300 µm distal to the apex of the lesser curvature of the aortic arch. Aortic root lesions were measured as the average of three sections separated by 100 µm beginning at the site of appearance of the coronary artery and valve leaflets. Atherosclerotic lesions were quantified from digitally captured images and OpenLab Software (Santa Clara, CA, USA), version 3.1.5. Additionally, a qualitative lesion score on a scale of 0 to 10 was assigned to each lesion, grading the histologic complexity and maturity of the atherosclerotic plaque as defined by criteria in Appendix A. 

### 4.5. VLDL Production Rate Experiments. 

After a 4 h fast, mice (five mice per group) were bled via the retro-orbital sinus and injected IV with 500 mg/kg body weight of Triton WR1339 (Sigma, Chemical Co) (15% *w*/*v* Triton WR1339 in 0.9% NaCl). Triglycerides (TG) in the plasma were measured before Triton injection and 90, 135, and 180 min post injection using Triglycerides-GB reagent kit (Roche Diagnostics, Indianapolis, IN, USA). VLDL triglyceride production rate is expressed as TG/min/mL plasma/gm liver [58]. Total plasma volume was approximated by multiplying body mass in grams by 0.0577. 

### 4.6. Statistical Analysis. 

Results are expressed as mean ± SEM. Statistical analysis was performed with StatView 5.0.1 software. The data were analyzed by one-way ANOVA. Values were deemed significantly different when *p* < 0.05.

## 5. Conclusions

In summary, these studies reveal that iNKT and other CD1d-expressing or CD1d restricted cells, which include type II NKT cells impact lipid and lipoprotein levels and the initiation and progression of atherosclerosis in a complex site-specific manner and in some cases gender specific manner. These studies suggest that endogenous and possibly microbial antigens related to the atherogenic process interact in complex ways with various NKT cell subsets. By directly comparing models with selective deficiencies in certain NKT cell subsets, we emphasize the unique contributions of these cell types to both lipid metabolism and atherogenesis. 

## Figures and Tables

**Figure 1 ijms-20-04566-f001:**
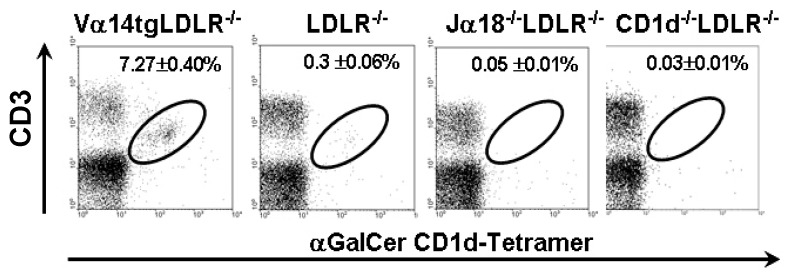
Flow cytometry analysis. Natural killer T (NKT) cell populations in the spleens from the indicated animal models fed Western type diet for 12 weeks were analyzed for CD3+ α-galactosyl ceramide (αGalCer) CD1d-Tetramer+ cells. *n* = 3–7 mice per group.

**Figure 2 ijms-20-04566-f002:**
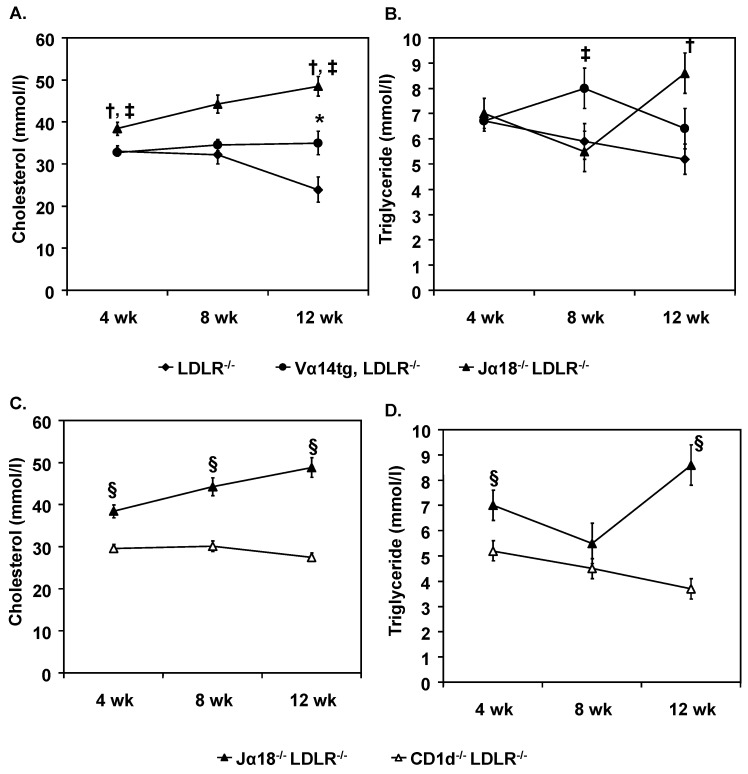
Plasma lipid levels. Plasma cholesterol (**A**,**C**) and triglyceride (**B**,**D**) in 4 h fasted plasma was measured every 4 weeks. Significance *p* < 0.005: * LDL receptor deficient (LDLR^−/−^) vs. Vα14tg/ LDLR^−/−^; † LDLR^−/−^ vs. Jα18^−/−^LDLR^−/−^; ‡ Vα14tg/LDLR^−/−^ vs. Jα18^−/−^LDLR^−/−^; §CD1d^−/−^LDLR^−/−^ vs. Jα18^−/−^LDLR^−/−^. For 4 weeks: *n* = 21–45. For 8 weeks; *n* = 12–34. For 12 weeks; *n* = 11–12.

**Figure 3 ijms-20-04566-f003:**
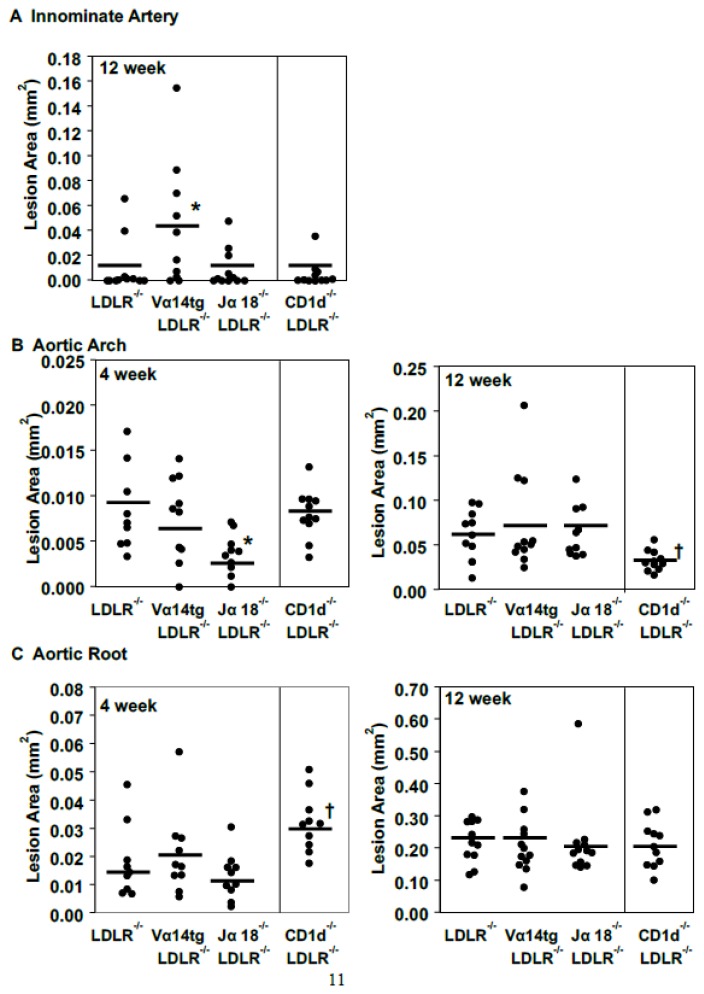
Atherosclerosis. Atherosclerosis was examined in the (**A**) innominate artery, (**B**) aortic arch, and (**C**) aortic root after 4 and 12 weeks on WTD. ^*^ Significant difference between Vα14tg/LDLR^−/−^ or Jα18^−/−^LDLR^−/−^ and LDLR^−/−^ mice. ^†^ Significant difference between Jα18^−/−^LDLR^−/−^ and CD1d^−/−^LDLR^−/−^ mice.

**Table 1 ijms-20-04566-t001:** Plasma lipoprotein cholesterol levels in female mice after 12-weeks of Western type diet (WTD) (*n*= 7–13). * *p* < 0.001 vs. LDLR^−/− †^
*p* < 0.0002 vs. Jα18^−/−^LDLR^−/− ‡^
*p* < 0.01 vs. Jα18^−/−^LDLR^−/−^.

	VLDLc	LDLc	HDLc
	mmol/L		
Vα14tg/LDLR^−/−^	17.7 ± 4.0	15.7 ± 1.5	2.4 ± 0.2
LDLR^−/−^	13.2 ± 2.4	12.3 ± 1.6	2.3 ± 0.3
Jα18^−/−^LDLR^−/−^	27.9 ± 2.7 *	18.6 ± 0.5 ^*^	2.5 ± 0.2

CD1d^−/−^LDLR^−/−^	12.2 ± 0.8^†^	14.4 ± 0.8^‡^	2.9 ± 0.4

**Table 2 ijms-20-04566-t002:** Hepatic triglyceride secretion. Female mice fed WTD for 12 weeks were fasted 4 h and injected with Triton WR1339 and plasma triglyceride levels in the plasma monitored over 3 h. *n* = 3–5 per group.

	Hepatic Triglyceride Secretion mg Triglyceride/dL min^−1^ mL Plasma^−1^ g Liver^−1^
Vα14tg, LDLR^−/−^	0.77 ± 0.33
Jα18^−/−^LDLR^−/−^	1.50 ± 0.65
CD1d^−/−^LDLR^−/−^	1.38 ± 0.30

**Table 3 ijms-20-04566-t003:** Histological lesion scores (see Appendix A) for female mice. IA = innominate artery, AA = ascending aorta, AR = aortic root. For 12 weeks of WTD, *n* = 11 or 12. For 4 weeks or WTD, *n* = 10 or 11.

	4 Weeks WTD		12 Weeks WTD		
	AA	AR	IA	AA	AR
Vα14tg, LDLR^−/−^	2.0 ± 0.2	1.8 ± 0.2	3.9 ± 0.7	4.7 ± 0.5	4.2 ± 0.3
LDLR^−/−^	2.1 ± 0.2	1.7 ± 0.2	1.7 ± 0.4	4.4 ± 0.5	4.5 ± 0.3
Jα18^−/−^LDLR^−/−^	1.8 ± 0.1	1.6 ± 0.1	1.4 ± 0.4	4.1 ± 0.3	3.5 ± 0.2
CD1d^−/−^LDLR^−/−^	2.2 ± 0.2	2.2 ± 0.2	2.0 ± 0.3	3.6 ± 0.3	4.3 ± 0.4

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
