# Peer review of "Invariant Natural Killer T-Cells and Total CD1d Restricted Cells Differentially Influence Lipid Metabolism and Atherosclerosis in Low Density Receptor Deficient Mice"

_ijms, 2019, doi:10.3390/ijms20184566_

Round 1

Reviewer 1 Report

Introduction:

line 38 where is written: related antigen presenting molecule CD1

should be written: related antigen presenting molecule CD1d

The results section of the manuscript should be improved.

Figure legend 1:

should be written Flow cytometry analysis and not Flow analysis.

The description of the results should be done in the text not in figure legend, so remove the results description from the legend.

In this figure should be removed the second panel (NK1.1 versus CD3) as it is not a reliable marker of NKT cells.

The results described in the paragraphs form line 107 to 123 should be described in figure(s) with statistics done. The graphics in the figures should be done with points representing individual data and not bars with means. Specially as for some conditions only two mice were analyzed. The authors are advise to include more subjects per group.

Figure S2 should also contain a representative plot of the flow cytometry gating.

Line 164 should be written CD3 and not CD3e. The sentence starting in line 164 and finishing in line 166 should be removed as NK1.1 versus CD3 is not a reliable marker of NKT cells.

In line 168 is written Fig. and in the rest of the manuscript is written Figure. The authors should use always the same form.

Author Response

Introduction:

line 38 where is written: related antigen presenting molecule CD1.  should be written: related antigen presenting molecule CD1d

Response:  Thank you.  This has been corrected.

The results section of the manuscript should be improved.

Figure legend 1:

should be written Flow cytometry analysis and not Flow analysis.

Response:  The requested change has been made

The description of the results should be done in the text not in figure legend, so remove the results description from the legend.

Response:  The information in the figure legend has been incorporated into the text (lines 78-83).

In this figure should be removed the second panel (NK1.1 versus CD3) as it is not a reliable marker of NKT cells.

Response:  The second panel has been removed and the figure legend revised.

The results described in the paragraphs form line 107 to 123 should be described in figure(s) with statistics done. The graphics in the figures should be done with points representing individual data and not bars with means. Specially as for some conditions only two mice were analyzed. The authors are advise to include more subjects per group.

Response:  The information about the statistics (SEM and p value) for the splenic analysis of Va14tg/LDLR-/- mice fed chow and WTD is now included in the text.  The statement about the LDLR-/- mice was deleted since the n was too low and the concluding statement can be made without that data. 

Figure S2 should also contain a representative plot of the flow cytometry gating.

Response:  Figure S2 was replotted with the individual data points.  We were not able to retrieve a flow plot for this data due to a computer crash; only the analysis was found on other computers.

Line 164 should be written CD3 and not CD3e. The sentence starting in line 164 and finishing in line 166 should be removed as NK1.1 versus CD3 is not a reliable marker of NKT cells.

Response: Both of the requested changes have been made in the text.

In line 168 is written Fig. and in the rest of the manuscript is written Figure. The authors should use always the same form.

Response:  We have made sure that Figure is used throughout the manuscript instead of the abbreviation.

Reviewer 2 Report

The conclusions of this manuscript are predicated upon the separate identity and presence of type I and type II NKT cells. However, the authors fail to satisfactorily establish the presence/absence of these cells populations in their animal models. While the presence of increased NKT cells in Va14tg mice is clear, the distribution of type I vs type II NKT cells is not appropriate. If 3% of splenocytes are NKT cells, how are 7% of the cells type I NKT cells? Furthermore, the differences between Ja18-/- and CD1d-/- are similarly inappropriate. The total % of NKT cells is the same in these 2 strains, dispite the fact that CD1d-/- mice should be missing 2 populations while Ja18-/- mice lack only 1. In order to appropriately classify these subsets, total NKT cells should be gated from splenocytes and then the disbribution of type I and type II determined from within this population. It is further recommended to gate out negative cells to increase the visibility of NKT cells within the total splenocyte population. Depending on this validity, the conclusions of the rest of the manuscript may change.

Secondly, the authors should read and apply the findings of Pasquet, et al. in their 2017 publication in JI entitled "Functionality of lung type II NKT cells developing in CD1d−/− animals." 

Author Response

The conclusions of this manuscript are predicated upon the separate identity and presence of type I and type II NKT cells. However, the authors fail to satisfactorily establish the presence/absence of these cells populations in their animal models. While the presence of increased NKT cells in Va14tg mice is clear, the distribution of type I vs type II NKT cells is not appropriate. If 3% of splenocytes are NKT cells, how are 7% of the cells type I NKT cells? Furthermore, the differences between Ja18-/- and CD1d-/- are similarly inappropriate. The total % of NKT cells is the same in these 2 strains dispite the fact that CD1d-/- mice should be missing 2 populations while Ja18-/- mice lack only 1. In order to appropriately classify these subsets, total NKT cells should be gated from splenocytes and then the disbribution of type I and type II determined from within this population. It is further recommended to gate out negative cells to increase the visibility of NKT cells within the total splenocyte population. Depending on this validity, the conclusions of the rest of the manuscript may change.

Response: Reviewer 1 rightfully indicates that CD3+NK1.1 cells represent more than NKT cells. Thus, exactly what markers we should use for total NKT cell gate in splenocytes is not apparent.  We can identify type I cells with the aGalCer-loaded CD1d tetramer (which we have done), but we did not have access to a sulfatide-loaded CD1d tetramer to also specifically measure type II cells. Given that we used the aGalCer-loaded Cd1d tetramer to measure iNKT cells in the Ja18-/- and CD1d-/- mice which are deficient in both strains, it is not surprising to us that the very low residual labeling is similar in the two strains.  

Secondly, the authors should read and apply the findings of Pasquet, et al. in their 2017 publication in JI entitled "Functionality of lung type II NKT cells developing in CD1d−/− animals.

Response:  The abstract has been cited in the manuscript on lines 280-283.

Round 2

Reviewer 2 Report

The authors still have not shown the presence of type I and type II NKT cells despite their focus on these two populations. For all of the transgenic mice that they use, they need to identify those 2 populations in order to believe that their conclusions are valid. Granted that there is not tetramer to stain all type II NKT cells, but this can be approximated by the following flow cytometric stains.

Type I NKT: lymphocyte+ single cell+ B220- CD3+ NK1.1+ aGalCer-Tetramer+

Type II NKT:  lymphocyte+ single cell+ B220- CD3+ NK1.1+ aGalCer-Tetramer-

If they use the gating strategy above and display the CD3+ gate with NK1.1 vs Tetramer, you can distinguish between type I and type II NKT cells. Because NKT cells vary in frequency in different tissues, it would be very relevant to not only analyze the spleen, but the liver as well due to its involvment in cholesterol metabolism. Only then can the conclusions drawn between the mouse strains used meaningfully distinguish between type I and type II NKT cells.

Author Response

We appreciate the reviewer's concern about concluding that type II NKT cells may be mediating the observed differences between Ja18-/- and CD1d-/- mice.  Indeed, we did not specifically measure the level of the type II NKT cells in the two strains. Since CD1d is expressed on other cell types (immune cells and nonparenchymal cells), the absence of CD1d in these cells may also have contributed to the phenotype differences observed.  As a result, we have revised the manuscript to indicate that in the CD1d-/- mice both CD1d expressing and CD1d restricted cells that include type II NKT cells are affected. Differences in any or all of these cells may be contributing to the differences observed in the two strains.  We hope that these modifications are acceptable to the reviewer.

See attached tracked revised manuscript.

Round 3

Reviewer 2 Report

It is apparent that the authors are not going to show the presence of type I vs type II. At least they have toned down their conclusions.